**www.cambridge.org/qrd**

# Sulfur-mediated chalcogen versus hydrogen bonds in proteins: a see-saw effect in the conformational space

Chalcogen bond; divalent sulfur; hydrogen bond; protein folding; protein secondary structure; sigma hole

**Corresponding author:**
Kayarat Saikrishnan;
Email: saikrishnan@iiserpune.ac.in

Vishal Annasaheb Adhav[1] , Sanket Satish Shelke[1] ,
Pananghat Balanarayan[2] and Kayarat Saikrishnan[1]

[1]Department of Biology, Indian Institute of Science Education and Research, Pune, India and [2]Department of Chemical Sciences, Indian Institute of Science Education and Research, Mohali, India

## Abstract

Divalent sulfur (S) forms a chalcogen bond (Ch-bond) *via* its σ-holes and a hydrogen bond (H-bond) *via* its lone pairs. The relevance of these interactions and their interplay for protein structure and function is unclear. Based on the analyses of the crystal structures of small organic/organo-metallic molecules and proteins and their molecular electrostatic surface potential, we show that the reciprocity of the substituent-dependent strength of the σ-holes and lone pairs correlates with the formation of either Ch-bond or H-bond. In proteins, cystines preferentially form Ch-bonds, metal-chelated cysteines form H-bonds, while methionines form either of them with comparable frequencies. This has implications for the positioning of these residues and their role in protein structure and function. Computational analyses reveal that the S-mediated interactions stabilise protein secondary structures by mechanisms such as helix capping and protecting free β-sheet edges by negative design. The study highlights the importance of S-mediated Ch-bond and H-bond for understanding protein folding and function, the development of improved strategies for protein/peptide structure prediction and design and structure-based drug discovery.

## Introduction

Non-covalent interactions are fundamental for protein folding, structural stability and function. Traditionally, hydrogen bonds (H-bonds), hydrophobic effects and electrostatic and van der Waals interactions are assumed to be the major drivers of protein folding and stability (Dill and MacCallum, 2012; Pace *et al.,* 2014). However, the essentiality of other weak interactions in sculpting protein structures is also being discovered. For example, the importance of weak H-bonds, cation/anion-π, $n \rightarrow \pi^\star$ and dihydrogen bonding (H···H) interactions for the stability of protein structures is well understood (Derewenda *et al.,* 1995; Gallivan and Dougherty, 1999; Manikandan and Ramakumar, 2004; Matta et al., 2003; Bartlett *et al.,* 2010; Lucas *et al.,* 2016; Forbes *et al.,* 2017; Newberry and Raines, 2019; Juanes *et al.,* 2020). Apart from C, O, N and H that form these non-covalent interactions, divalent sulfur (S) is present in methionines (Met-S$^\delta$), cysteines (Cys-S$^\gamma$) and cystines of proteins. S has unique bonding properties that allow it to interact with both electrophiles and nucleophiles (Rosenfield *et al.,* 1977; Guru Row and Parthasarathy, 1981). Consequently, methionine, cystine and cysteine are expected to contribute to distinct polar interactions in proteins, making it imperative to study them to better understand their structural properties and folding.

Although the polar bonding properties of S have long been known (Fig. 1a) (Rosenfield *et al.,* 1977; Guru Row and Parthasarathy, 1981), it has gained prominence in the field of organic molecules over the last decade (Andersen *et al.,* 2014; Beno *et al.,* 2015; Rao Mundlapati *et al.,* 2015; Pascoe *et al.,* 2017; Wang and Fujii, 2017; Motherwell *et al.,* 2018; Riwar *et al.,* 2018; Scilabra *et al.,* 2019; Haberhauer and Gleiter, 2020; Kolb *et al.,* 2020; Wang *et al.,* 2020). In contrast, these properties of S are often overlooked in the study of proteins. Many standard textbooks of biochemistry categorise methionine as a non-polar hydrophobic amino acid (Lehninger *et al.,* 2000; Berg *et al.,* 2002). However, the lone pairs of S can interact with electrophiles, such as H atoms, to form H-bonds with donor O or N (Fig. 1a) (Zhou *et al.,* 2009; Rao Mundlapati *et al.,* 2015; Chand *et al.,* 2020). Although S is less electronegative than O/N, the strength of S-mediated H-bonds such as N–H···S is comparable to the conventional N–H···O bond mainly because of its larger size, higher polarisability and the diffuse electron cloud (Gregoret *et al.,* 1991; Rao Mundlapati *et al.,* 2015). Additionally, the divalent S has two electropositive regions on the extension of its two covalent bonds, referred to as σ-holes, which can interact with various nucleophiles (Fig. 1a) (Murray *et al.,* 2012; Politzer *et al.,* 2014; Politzer *et al.,* 2017). The interaction made by a σ-hole of S with a nucleophile is categorised as a chalcogen bond (Ch-bond) (Aakeroy *et al.,* 2019).

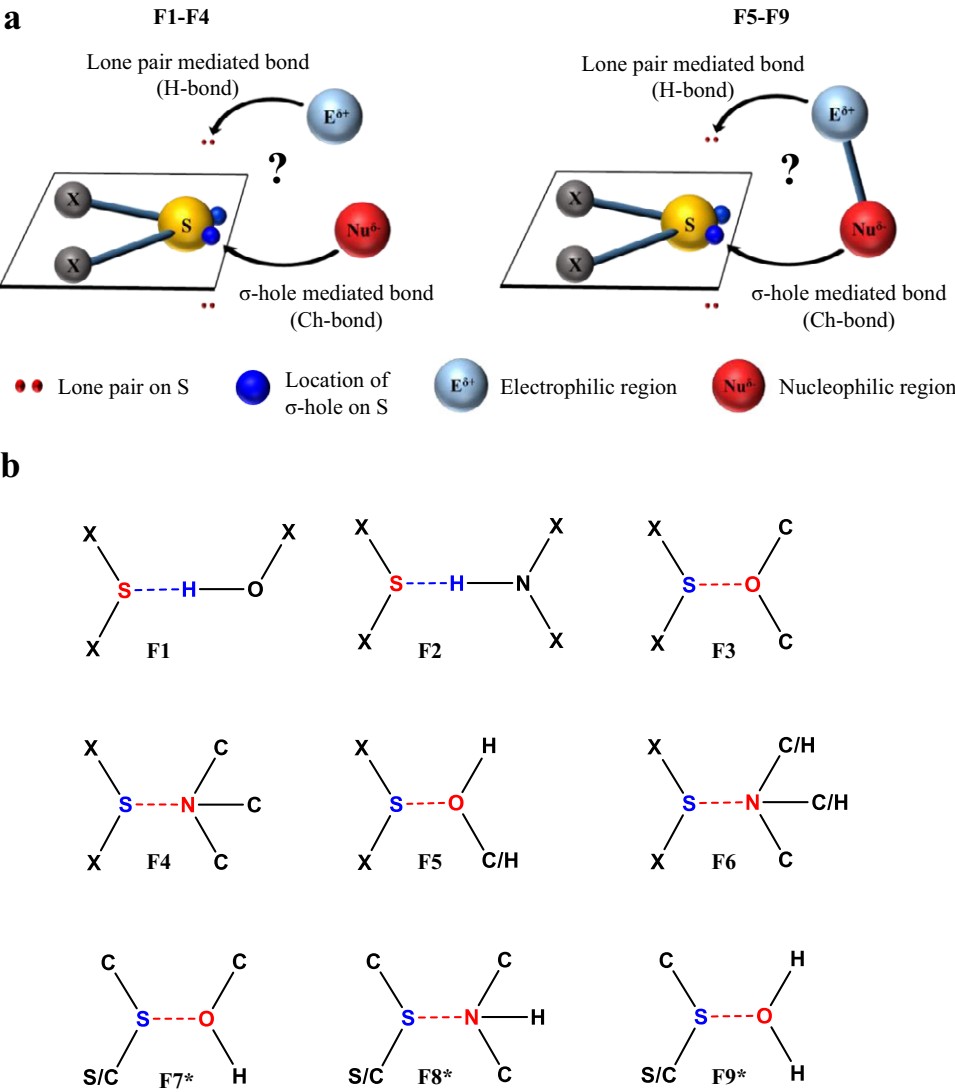

**Figure 1.** An approach of electrophiles and nucleophiles towards divalent S. (*a*) Approach of an electrophile or a nucleophile (upper panel) and a covalently linked electrophile–nucleophile fragment (lower panel) towards S. (*b*) Fragments analysed using the Cambridge Structural Database and the Protein Data Bank (PDB). *Fragments are from protein structures in the PDB.

Ch-bonds have been shown to play important roles in the self-assembly and catalysis of organic molecules (Iwaoka *et al.,* 2002; Benz *et al.,* 2017; Mahmudov *et al.,* 2017; Chen *et al.,* 2018; Lim and Beer, 2018; Vogel *et al.,* 2019; Carugo *et al.,* 2021; Jena *et al.,* 2022). Ch-bonds have a specific directionality, and a recent spectroscopic study on thiophenes has shown that the bond can be as strong as a conventional H-bond (Pascoe *et al.,* 2017). However, unlike the latter, the strength of a Ch-bond is independent of solvent polarity (Pascoe *et al.,* 2017). The occurrence of a divalent S-mediated Ch-bond in proteins has been documented previously and hypothesised to be functionally significant (Pal and Chakrabarti, 2001; Iwaoka *et al.,* 2002; Iwaoka and Isozumi, 2012, 2006; Iwaoka and Babe, 2015). Despite its potential importance, the precise role of the Ch-bond in protein structures and its effect on protein stability have remained unaddressed.

Many functional groups in proteins contain both electrophilic and nucleophilic centres with which S can interact to form H- or Ch-bonds (Fig. 1*a*). Hence, the direction of approach of the functional groups with respect to S and the nature of the bond

formed are interlinked and could influence protein conformation. This prompted us also to ask whether S forms an H-bond or a Ch-bond with a functional group having both electrophilic and nucleophilic centres and what determines the choice between them (Fig. 1*a*). Here, we have addressed the above questions through extensive computational, cheminformatics and bioinformatics analyses. The study reveals the importance of Ch-bonds in the stability of protein structures. Using potential energy surface (PES) scan and atoms in molecules (AIM) analysis, we show that both H- and Ch-bonds can augment the stability of protein secondary structures, and through bioinformatics analyses, we identify some of the mechanisms of fold stabilisation. Analysis of the structures in the Cambridge Structural Database (CSD) and the Protein Data Bank (PDB), and molecular electrostatic surface potential (MESP) calculations reveal the factors that influence the choice between H-bond and Ch-bond by S. The study, therefore, unravels the underappreciated role of S-mediated interactions, particularly Ch-bonds, in protein structure, stability and function.

## Methods

### Computational methods

All monomer and dimeric complexes used in this study were opti-mised at the M06/6-311++G(3DF,3PD) level using the Gaussian09 programme (Zhao and Truhlar, 2008; Frisch *et al.*, 2009). The positive eigenvalues of the Hessian evaluation confirmed that these optimised complexes are minima on the PES. The $(CH_3)_2S:OH_2$ and $(CH_3)_2S:NH_3$ complexes were used to investigate energetically favourable regions for S⋯H–O/N interaction. Similar investigations were per-formed on $Cl(CH_3)S:OH_2$ for S⋯O and $Cl(CH_3)S:NH_3$ for S⋯N interactions. To carry out a spherical energy scan using these com-plexes, $d$ (distance between S and H/O/N) was kept constant, whereas $\theta$ (the angle between the centroid (c) of a triangle defined by C–S–C/Cl, S and H/O/N) and $\delta$ (the torsion angle between C, $c$, S and H/O/N) were varied in steps of 2˚. $\delta$ was varied from 90˚ to −90˚ and $\theta$ from 60˚ to 178˚. Complexation energies ($\Delta Es$) for different $\theta$ and $\delta$ values without basis set superposition error correction were calculated for all these complexes (Hobza and Řezáč, 2016). MESP topographical analyses characterising the strength of lone pairs ($V_{min}$) of divalent S were carried out using the rapid topography mapping implemented in DAMQT (Yeole *et al.*, 2012; Kumar *et al.*, 2015). Texturing of MESP of molecules at defined density surface ($V_{S,max}$) was carried out using Gaussview 5.0 (Dennington *et al.*, 2009). Model systems used to investigate the role of S⋯O interactions in α-helices and a β-strands were from the PDB coordinates of 1PVH and 4KT1. All the side chain atoms in these fragments were deleted, and the $C^\beta$ atom was replaced by H. A partial energy minimisation was carried out where all atoms other than hydrogen were frozen. The torsion angles ($\chi$) between N, $C^\alpha$, $C^\beta$ and S were varied for the PES scan. AIM analysis was carried out using AIM2000 (König *et al.*, 2001).

### CSD analysis

Fragments F1–F6 provided in Fig. 1*b* were queried in structural data retrieved from CSD 5.43 (Groom and Allen, 2014) (version 5.43 with updates till March 2022) using ConQuest version 2022 1.0 (Bruno *et al.*, 2002a). The following criteria of ConQuest were used for the search: (1) only intermolecular contacts; (2) 3D coordinates determined for all the atoms; (3) structures with crystallographic *R-factor* ≤10%; (4) no disorder in crystallographic data; (5) no error in 3D atomic coordinates; (6) no polymeric structures; and (7) nor-malisation of terminal H position. Data thus obtained were further processed and analysed using Mercury 2022 1.0 (Bruno *et al.*, 2002b).

Searches were done for only intermolecular contacts using the criteria $d_{S\cdots O} \leq 3.32$ Å and $d_{S\cdots N} \leq 3.35$ Å. For the identification of H-bond, $d_{S\cdots H} \leq 2.8$ Å, which is 0.2 Å shorter than the sum of the van der Waal radii of S and H (Bondi, 1964), was chosen for higher stringency. The number of bonded atoms to S in F1–F6 were 2. The number of bonded atoms to O and N in F3–F6 were 2 and 3, respectively. The number of such fragments in CSD is provided in Table 1. The approach of H or O/N towards S in space was investigated using the 3D parameters provided in ConQuest (Fig. 2*a*). To segregate H- and Ch-bonds based on their $\theta$ and $\delta$ values, we calculated the mean values for clusters in F1 and F3. The range of $\theta$ and $\delta$ defining H- and Ch-bonds about the respective mean values were obtained by taking their mean ± standard devi-ation at 1 sigma of the calculated values. The values for the limits were rounded off to the closest value that was a multiple of 5. The mean of the angular values of $\delta$ was calculated using their modulus. This angular range of $\theta$ and $\delta$ for H- and Ch-bonds thus obtained

**Table 1.** A summary of CSD and PDB analysis

| Fragment | Contact | $N_T$ | $N_C$ (%) |
|---|---|---|---|
| F1 | S⋯H–O | 9,603 | 609 (6.34) |
| F2 | S⋯H–N | 11,497 | 1,647 (14.33) |
| F3 | S⋯O | 13,711 | 474 (3.46) |
| F4 | S⋯N | 9,466 | 31 (0.38) |
| F5 | S⋯O | 9,293 | 644 (6.92) |
| F6 | S⋯N | 8,173 | 119 (1.46) |
| F7[a] | S⋯O | N.D. | 1,320 (methionine) 285 (cystine) |
| F8[a] | S⋯N | N.D. | 1,005 (methionine) 228 (cystine) |
| F9[a] | S⋯O | N.D. | 8,737 (methionine) 980 (cystine) |

*Note: $N_T$ is the total number of independent pairs of fragments found in the CSD structures. $N_C$ is the total number of S⋯O or S⋯H–O and S⋯N or S⋯H–N contacts found in the CSD and the PDB, having distance between S and the atom less than the sum of their van der Waals radii. The values in the parentheses are equal to $(N_C/N_T) \times 100$ and represent the frequency of occurrence of the above-mentioned contacts in the CSD.*
*Abbreviations: CSD, Cambridge Structural Database; PDB, Protein Data Bank.*
*[a]Fragments are from protein structures in the PDB.*

were used throughout the study. All the plots and figures were generated using OriginPro 9.0 (Seifert, 2014).

### PDB analyses

Protein structures determined using X-ray crystallography in the PDB (Rose *et al.*, 2017) were downloaded in July 2022. A set of protein structures with resolution ≤2.0 Å, *R-factor* ≤25% and pairwise sequence identity ≤90% was generated using the PISCES server (Wang and Dunbrack, 2005), resulting in 25,107 structures. The criterion for pairwise sequence identity was excluded for searching ligands containing aromatic S. These structures were analysed using an in-house script written in Python 3.7.1. Search for H-bonds and Ch-bonds was made using the criterion of $d$ (Å) ≤ (sum of van der Waals radii of S and O/N). To minimise the effect of structural constraints on the direction parameters, we excluded the contacts where S and O/N were separated by less than seven covalent bonds or were intramolecular. The direction criteria obtained using the CSD analysis were used to distinguish between H- and Ch-bonds.

To understand the effects of H- and Ch-bonds on proteins' secondary structures, we searched the PDB for S⋯H–N ($d_{S\cdots N} \leq 3.6$ Å, relaxed from 3.35 Å because it usually peaks at 3.6 Å in proteins (Zhou *et al.*, 2009)) and S⋯O ($d_{S\cdots O} \leq 3.32$ Å) bonds made by the peptide backbone with S. An additional criter-ion of $120˚ \leq \zeta \leq 240˚$, where $\zeta$ is a torsion angle (see Supplementary Fig. 1 for more details), was applied to exclude structures where N–H was not pointing towards the lone pair regions of S. Secondary structure information was obtained from the header in the PDB files. For *N*-terminal capping, we searched for S⋯H–N bonds where the amino group was of $N_1$, $N_2$ or $N_3$ residue at the N-terminus of the helix. While for C-terminal capping, we searched for S⋯O interactions where the carbonyl O was of $C_1$, $C_2$ or $C_3$ residue at the C-terminus of the helix (Aurora and Rose, 1998). A similar analysis was performed to investigate the effects of S⋯H–N, S⋯O and S⋯N interactions on the stability of α-helices and β-strands. In this case, the last four residues at the N- and C-termini of the helix

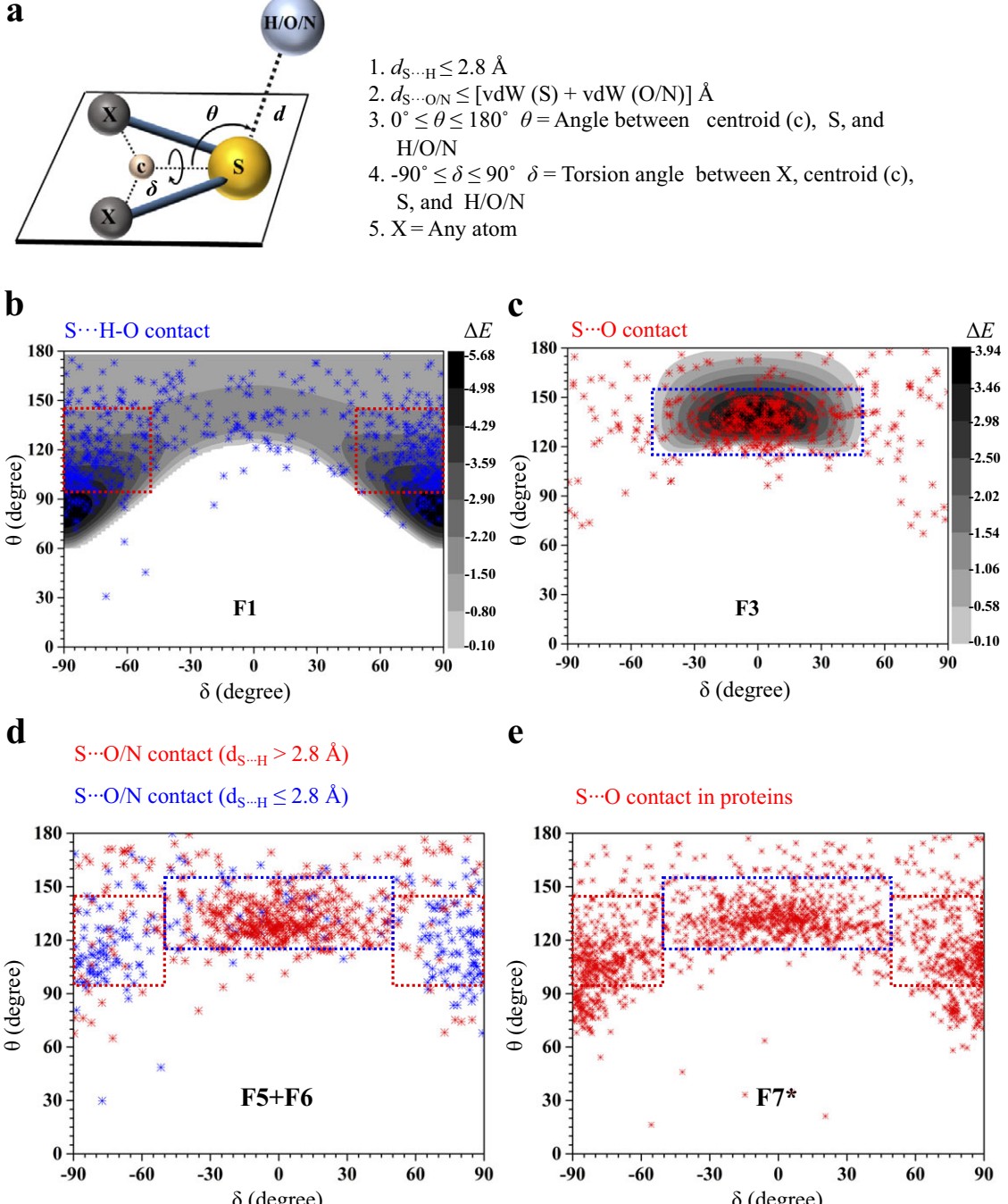

**Figure 2.** Nature of non-covalent interactions formed by S. (*a*) Definition of geometrical parameters *d*, *θ* and *δ*. (*b*) Mapping of the *θ* and *δ* values of S···H–O contacts (blue dots) in F1 with computationally calculated Δ*E*s in the background (greyscale). The $(CH_3)_2S:OH_2$ complex was used as a model system to calculate Δ*E*s for F1; (*c*) S···O contacts (red dots) in F3. The $Cl(CH_3)S:O(CH_3)_2$ complex was used as the model system to calculate Δ*E*s in the background (greyscale). (*d*) S···O/N contacts with $d_{S···H} > 2.8$ Å in red and those with $d_{S···H} ≤ 2.8$ Å in blue. (*e*) S···O contacts formed by methionine and cystine in F7. The boxes shown in the figures represent the statistically obtained favourable range for *θ* and *δ* values for H- or Ch-bonds.

were excluded, whereas all the residues belonging to the strand were considered. Search for metal-chelating cysteine used a distance between S and the metal to be within 1.9–2.8 Å, which ensured that interactions with metals of different ionic radii were identified. This distance range was calculated using an in-house script for all interactions between S and metals in protein structures in the PDB. Figures of protein structures were made using Chimera 1.13.1 (Pettersen *et al.,* 2004).

## Results and discussion

### *Geometrical features that distinguish S-mediated Ch-bond from H-bond*

Structures in the PDB solved using X-ray crystallography often lack positional information of H atoms, making it difficult to identify if the non-covalent bond between S and O/N is H-bond or Ch-bond. We devised a methodology to distinguish between the H- and

Ch-bonds even if positional information of the H atom was absent. Towards this, we first analysed fragments F1 and F2 in the CSD having positional information of H to identify the preferred direction of approach of H towards S to form H-bond (Fig. 1b). Fragments F3 and F4 were studied to characterise the preferred direction of approach of O/N towards S to form Ch-bond. $\theta$ and $\delta$ were used to characterise the angular distribution for H- and Ch-bonds (Fig. 2a). The parameters measured for all such contacts in the CSD were plotted to illustrate the preferred range of distances and angles of these interactions (Fig. 2b,c and Supplementary Fig. 2a,b). The angular distribution thus obtained was compared with the complexation energy $\Delta E$ obtained from PES scans at different values of $\theta$ and $\delta$ for the model systems (Fig. 2b,c).

Two distinct clusters were observed in the $\theta$–$\delta$ plot for S···H–O contacts. The boundary values of the two clusters were (i) $95° \leq \theta \leq 145°$ and $-90° \leq \delta < -50°$ and (ii) $95° \leq \theta \leq 145°$ and $50° < \delta \leq 90°$, respectively. This matched with the location of the PES scan minima (Fig. 2b and Supplementary Fig. 2a). The clusters represented the direction of approach of the electrophile towards the lone pairs of S (Supplementary Fig. 3a). In the case of S···O contacts, a single cluster was observed at a different region of the $\theta$–$\delta$ plot ($115° \leq \theta \leq 155°$ and $-50° \leq \delta \leq 50°$), which overlapped with the PES scan minimum (Fig. 2c and Supplementary Fig. 2b). The direction corresponded to the approach of the nucleophile towards the σ-hole on S (Supplementary Fig. 3a; Politzer et al., 2013; Aakeroy et al., 2019). Outliers in the plots were due to other strong interactions within the molecules, such as other H-bonds and stacking interactions (Supplementary Fig. 3b–d). The number of S···N interactions was much less than S···O (Table 1) possibly because of N being conjugated in most of the structures resulting in the lack of lone pair electrons for the formation of Ch-bond.

### Delineation of Ch-bond from H-bond in groups having electrophilic and nucleophilic centres

We next sought the nature of bonding between S and functional groups having both electrophilic and nucleophilic centres because they are common in proteins. For this, we studied fragments F5 and F6 in the CSD (Fig. 1a,b and Table 1). Using distance criteria, we ensured that these fragments formed either S···H–O/N or S···O/N interaction. Due to structural constraints, both interactions did not occur simultaneously. From this set of interactions, contacts satisfying $d_{S···H} \leq 2.8$ Å were assigned as H-bond and the rest as Ch-bond. Note that this filtering strategy excluded those H-bonds having a distance between S and O/N greater than 2.8 Å. The CSD analysis revealed three clusters in the $\theta$–$\delta$ plot (Fig. 2d). Interactions in two of these clusters had $\theta$–$\delta$ values expected for H-bond and most of them satisfied the criterion $d_{S···H} \leq 2.8$ Å (Fig. 2b). S···O/N interactions with $d_{S···H} > 2.8$ Å primarily clustered with $\theta$–$\delta$ distribution that matched the directionality of Ch-bond (Fig. 2c). A few interactions in this cluster had $d_{S···H} \leq 2.8$ Å. Note that H–O/N groups that formed Ch-bond with S could also form H-bond with a neighbouring acceptor atom (Supplementary Fig. 3a).

In proteins, Ch- or H-bond can be formed between methionine and cystine with side chains of serine, threonine or tyrosine, or the backbone amide or water (fragments F7–F9) is equivalent to fragments F5 and F6. As there were very few examples of $X_1$–S–H in the CSD analysis discussed above, we excluded interactions made by free cysteine from the analysis of PDB

structures. As in the case of fragments F5 and F6, the $\theta$–$\delta$ plot obtained by analysing F1–F4 revealed the segregation of angular values into three clusters corresponding to either H-bond or Ch-bond (Fig. 2e and Supplementary Figs 2c,d and 4). In summary, the $\theta$–$\delta$ plot allowed us to identify and distinguish Ch-bond from H-bond in protein structures without positional information of H atoms.

### Electronic environment of S determines the choice between Ch- and H-bond

We next sought to find what dictated the choice between the formation of Ch-bond and H-bond. In general, the formation of H- or Ch-bond depends on the strength of lone pairs and σ-holes on S, respectively, which are in turn affected by the nature of substituent groups (Adhikari and Scheiner, 2014; Kumar et al., 2014). Using MESP, we found the same in model systems relevant to biomolecules (Fig. 3a,b). The MESP analysis revealed two $V_{min}$ (MESP minimum), corresponding to the lone pairs on S in all the model systems (Fig. 3a). The electrostatic potential maps also showed the presence of two σ-holes on the extension of the S–X bonds, except in the case of $[Fe(SCH_3)_4]$ complex (Fig. 3b) because of the anionic nature of metal-chelated S (Hirano et al., 2016). Interestingly, we noted the ability of substituents to modulate the strength of the lone pairs and σ-holes on S in a reciprocal manner, which consequently was expected to affect the nature of the bond formed.

To see if this was true, we categorised all the contacts obtained from CSD based on the substituents linked to S (Fig. 4a and Supplementary Table 1). We analysed the corresponding structures to check if S formed Ch-bond or H-bond. Eighty-eight percent of S(Ar) and 73% of E–S–Y formed Ch-bond. In sharp contrast, more than 97% of M–S–Y formed H-bond. In comparison, saturated C/S/H substituents (R–S–R) appeared to have a lesser influence on the choice of the bond formed. The number of H-bonds (52%) was almost comparable to the number of Ch-bonds (48%) (Supplementary Table 1). The observations matched the expectations from the MESP analysis performed on the model systems. In summary, divalent S, when part of an aromatic ring or bonded to an electron-withdrawing group, is most likely to form a Ch-bond, whereas S coordinated with a metal can form an H- but not a Ch-bond.

### Disulphide-linked S preferentially forms Ch-bonds, whereas metal-chelated cysteine form H-bonds in proteins

We studied interactions made by S of methionine or cystine with the hydroxyl, amino or carbonyl group of backbone amide, side chains of serine, threonine, tyrosine, aspartate, glutamate, arginine, lysine, histidine, asparagine, glutamine, tryptophan and bound water (Supplementary Table 2). Our analysis revealed that the disulphide-linked Cys-$S^\gamma$ was more frequently involved in Ch-bonds (85%) than H-bonds (15%). In comparison, Met-$S^\delta$ appeared to form H-bonds (57%) only marginally more than Ch-bonds (43%) (Fig. 4b and Supplementary Table 3). The MESP analysis showed that S bonded to two methyl groups (C–S–C), as in methionine, had comparable values of $V_{min}$ and $V_{S,max}$ (Fig. 3a, b). In contrast, $V_{S,max}$ on a disulphide-linked S (C–S–S) was larger than $V_{min}$, thus providing a rationale for cystine to preferentially form Ch-bonds.

Aromatic S preferentially formed Ch-bonds with groups containing O. This observation was consistent with previous reports

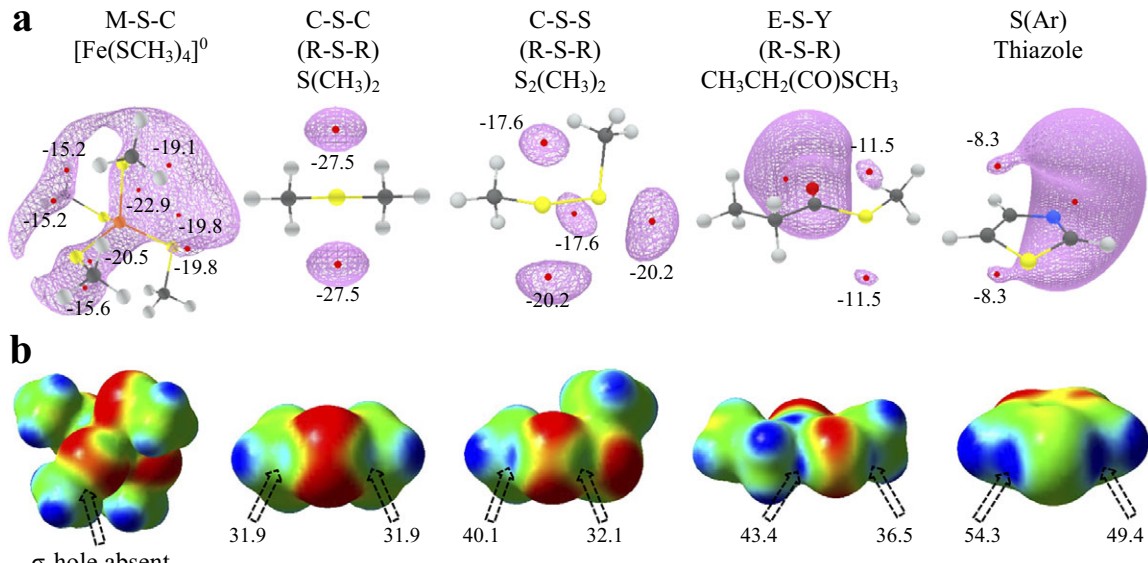

**Figure 3.** Effect of substitution on the electronic environment of S. (*a*) Molecular electrostatic surface potential (MESP) minimum values, $V_{min}$, in kcal mol$^{-1}$ represents the lone pair regions of the S-containing monomers used in this study. (*b*) MESP map of the monomers with the colour-coding range from $-6.28$ (red) to 43.93 kcal mol$^{-1}$ (blue) and textured on a 0.01 au density isosurface. The two σ-holes marked by arrows and their magnitude, $V_{S,max}$, in kcal mol$^{-1}$.

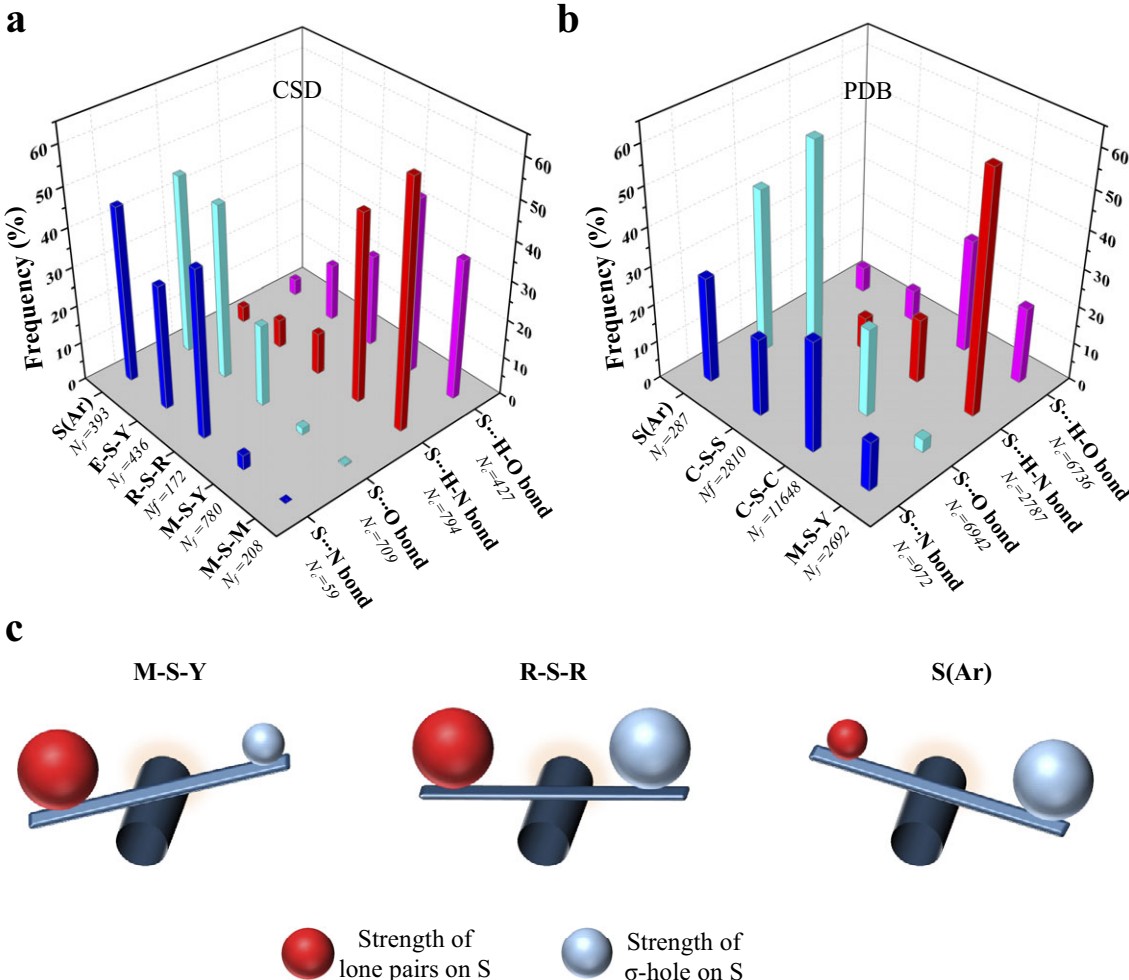

**Figure 4.** Rules for the formation of H- and Ch-bonds. (*a*) Histogram showing the frequency formation of Ch-bond and H-bond in the Cambridge Structural Database and (*b*) the Protein Data Bank with different electronic environments of S. M = any metal; Y = any element except M; R = saturated C, H and S; E = any electron-withdrawing group and S(Ar) = Aromatic S. (*c*) Substituent-dependent see-saw change in the strength of the lone pairs and σ-holes on S.

of Ch-bonds between S in the aromatic rings of drugs containing thiophene, thiazole and thiadiazole groups and O in target proteins (Thomas *et al.*, 2015; Zhang *et al.*, 2015; Koebel *et al.*, 2016; Kristian *et al.*, 2018). S-mediated interactions with functional groups containing N did not show these features presumably because the delocalised lone pairs of N in backbone amide or the side chain precluded the formation of Ch-bonds. We next analysed the PDB for non-covalent interactions formed by metal-chelated cysteines, which occur in many metalloproteins. Consistent with the rules stated above and independent of the identity of the metal, the thiolate of cysteine preferentially formed H-bonds (M–S–Y in Fig. 4*a*). In summary, our analyses of the structures in the CSD and the PDB revealed that the nature of the interaction between functional groups containing electrophilic and nucleophilic centres and S was influenced by the substituent-dependent see-saw change in the strength of the lone pairs and σ-holes on S (Fig. 4*b*).

### Role of S in helix capping

Capping satisfies the H-bond-forming abilities of the free backbone N–H or C=O of the terminal residues of an α-helix and is essential for the stability of α-helices in proteins and peptides (Aurora and Rose, 1998). The role of polar side chains of serine, threonine and asparagine, the acidic side chain of aspartate, the backbone amide of a neighbouring residue and metal-chelated S of cysteine in helix capping are well documented (Doig and Baldwin, 1995; Aurora and Rose, 1998), but not those of methionine and cystine. As the N-terminus and C-terminus of α-helices have free backbone N–H (electrophile) and free backbone C=O (nucleophile), respectively, we asked if Met-S$^\delta$ or Cys-S$^\gamma$ would interact and cap them.

We analysed protein structures in the PDB and found a number of examples of Met-S$^\delta$ or Cys-S$^\gamma$ interacting with backbone amino at the N-terminus or backbone carbonyl at the C-terminus of α-helix (Fig. 5*a* and Supplementary Table 4). Consistent with previous observations (Doig and Baldwin, 1995), we found that metal-chelated thiolates capped only the N-terminus of the helix by H-bonds. In contrast, 75% of Cys-S$^\gamma$ capped the C-terminus by Ch-bonds, whereas the remaining 25% capped the N-terminus by H-bonds (Fig. 5*b*). Amongst the examples involving Met-S$^\delta$, 37% of the interactions were H-bonds with backbone N–H of the N-terminal residues and 63% Ch-bonds with backbone C=O of the C-terminal residues (Fig. 5*c*).

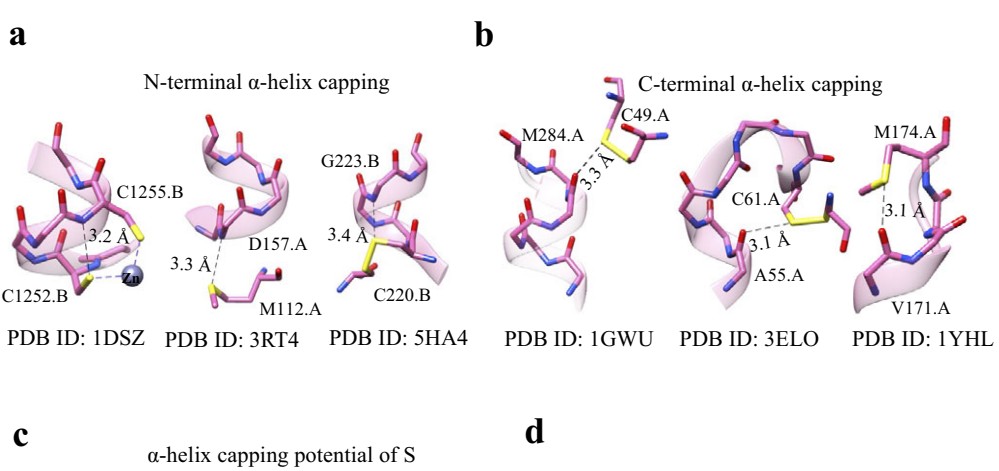

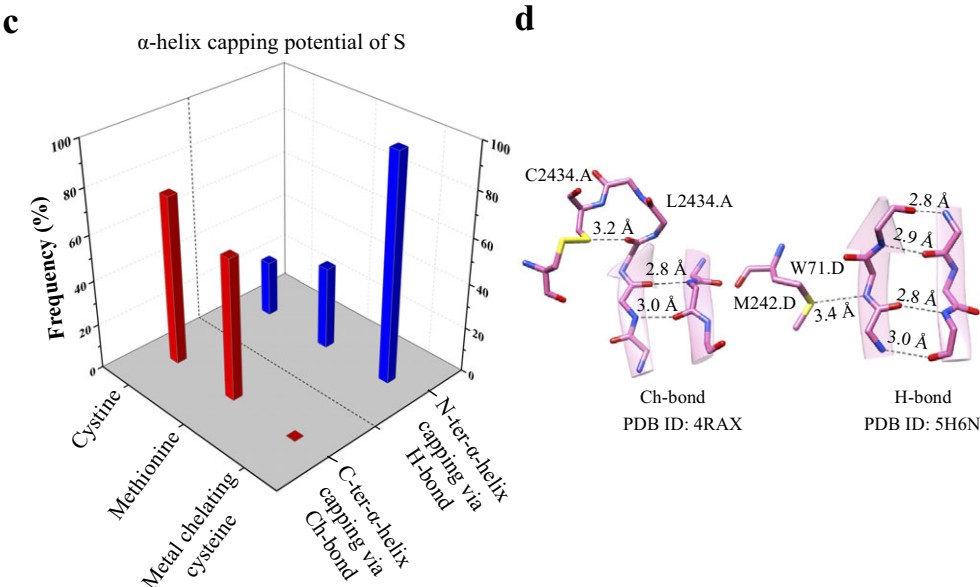

**Figure 5.** Helix capping and edge strand stabilisation. (*a*) Representative examples of H-bond capping the N-terminus of α-helices and (*b*) Ch-bond capping the C-terminus of α-helices. (*c*) Histogram showing the frequency of H-bond and Ch-bond interactions capping the N- and C-termini of α-helices by metal-chelating cysteine, methionine and cystine. (*d*) Representative examples of negative design involving Ch-bond and H-bond.

## Augmentation of the stability of regular secondary structures by S

In addition to backbone H-bonds, other non-covalent interactions such as C–H···O and $n \rightarrow \pi^*$ are important for the structural stability of α-helices or β-sheets (Derewenda *et al.,* 1995; Manikandan and Ramakumar, 2004; Bartlett *et al.,* 2010). An earlier study reported methionine-forming intra-helical and inter-strand Ch-bonds with backbone O (Pal and Chakrabarti, 2001). This prompted us to find if S could contribute to the stability of α-helices and β-strands through H- and Ch-bonds. We analysed structures in the PDB for Ch- and H-bonds between Met-$S^\delta$ or Cys-$S^\gamma$ with backbone O or N–H of residues of α-helices or β-strands (Supplementary Fig. 5 and Table 5). Interestingly, we found an S-mediated H-bond involving backbone N–H of β-strands at the edge of β-sheets (Fig. 5*d*). We also found Ch-bond formed by S with free backbone C=O of edge strands.

Many elements of negative design, a mechanism that prevents the β-strand dimerisation and stabilises an edge strand of β-sheets, have been documented previously (Richardson and Richardson, 2002; Koga *et al.,* 2012). This includes the interaction of other regions of the protein with the edge β-strand, disruption of backbone H-bond formation by proline or a β-bulge or use of inward-pointing charged residues to prevent strand-mediated dimerisation. Our analysis revealed that H-bond or Ch-bond formed by backbone N–H or C=O of edge β-strand, respectively, with a neighbouring Met-$S^\delta$ or Cys-$S^\gamma$, is another element of negative-design that can stabilise β-sheets. Additionally, we found that in some proteins, the free backbone N–H of the insertion residue of a classical β-bulge formed an H-bond with Met-$S^\delta$ located two residues ahead (Supplementary Fig. 5).

To gauge the potential contribution of Ch-bonds to stabilise regular secondary structures, we performed a PES scan by varying the torsion angle χ about the covalent bond between $C^\alpha$ and $C^\beta$ of the cystines whose S formed a Ch-bond. One fragment each from an α-helix and a β-strand were chosen for the calculations (Fig. 6*a*). A plot of relative conformational energy versus χ showed that the minimum conformational energy corresponded to the χ of the respective crystal structures (Fig. 6*b*). The AIM analysis for minimum energy conformations showed the presence of a bond critical point (BCP) between S and O. $\rho$-values of these BCPs were 0.007 au for 1PVH and 0.011 au for 4KT1 (Fig. 6*b*), which were in the range suggested previously for favourable non-covalent interactions, that is, 0.002–0.035 au (Bader, 1991). The analysis, thus, strongly suggested that Ch-bonds could provide extra stability to a particular conformation in protein molecules.

## Conclusions

In this study, we have tried to understand the role of H- and Ch-bonds formed by divalent S in proteins. Computational analyses showed that the S-mediated interactions contributed to the stability of protein conformation and secondary structures. Hence, we conclude that S-mediated Ch- and H-bonds, like other weak interactions, are an essential aspect of the energy landscape in protein folding that compensates for unfavourable conformational entropy changes through favourable interactions (Grantcharova *et al.,* 2001; Dobson, 2003). Furthermore, we envisage that cooperativity among S-mediated and other weak interactions is likely to modulate their strengths with direct implications for protein function, which remains to be studied (Adhav et al., 2022). For example, we speculate that the propensity and strength of Ch-bonds would increase upon

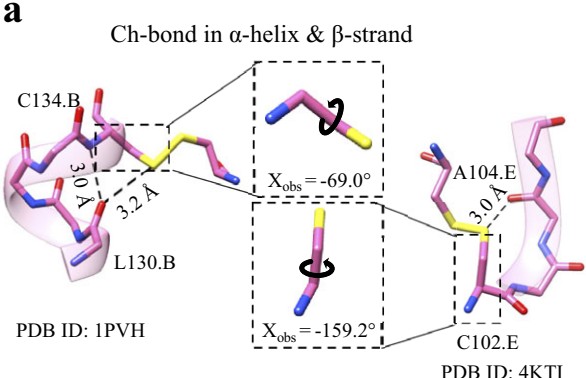

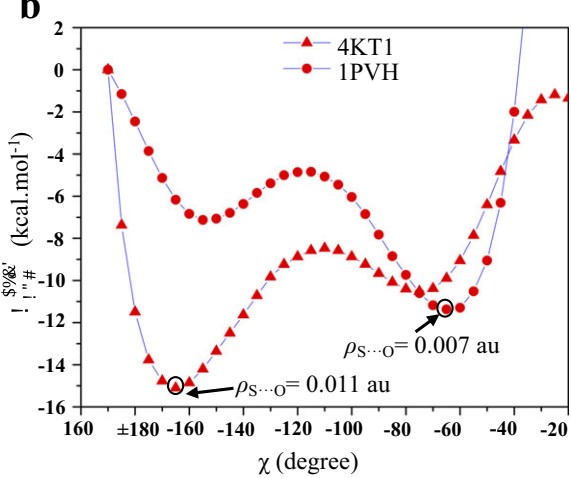

**Figure 6.** Stabilisation of α-helices and β-sheets by S. (*a*) Representative examples of Ch-bonds found in α-helical and β-sheet regions. (*b*) The plot for conformational energies as a function of χ (*E* at χ = 170˚ was assigned as 0.0 kcal mol⁻¹). Values of $\rho$ at bond critical points for the S···O interaction for the most energetically favourable structures are given in atomic units.

the delocalisation of lone-pair electron density of Cys-$S^\gamma$ to form an $n \rightarrow \pi^*$ interaction with a vicinal carbonyl group (Kilgore and Raines, 2018). Also, the computational analyses reported here do not delineate the contribution of hydrophobic and van der Waals interactions from those of the polar Ch-bond and H-bond interactions towards structural stability.

S-mediated Ch- and H-bonds can contribute to the structural stability and substrate specificity of proteins, like other interactions formed by polar amino acids. However, S-mediated interactions can have properties different from other polar non-covalent interactions, for instance, the resistance of Ch-bond strength to solvent polarity (Pascoe *et al.,* 2017), thus bringing additional diversity to the repertoire of weak interactions essential for biomolecular functions. This could be a reason why, despite their high biosynthetic cost (Doig, 2017), nature selected S-containing amino acids as part of the 20 building blocks of proteins. The wide variety of functionally relevant interactions made by S in proteins necessitates that these non-covalent interactions too are considered in the energy functions used for determining protein structures, folding pathways and binding properties. Also, the design and engineering of proteins and peptides would benefit from a better understanding of the distinct bonding properties of methionine and cysteine/cystine.

**Open peer review.** To view the open peer review materials for this article, please visit http://doi.org/10.1017/qrd.2023.3.

**Supplementary material.** The supplementary material for this article can be found at http://doi.org/10.1017/qrd.2023.3.

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
