## [Reviewer Report]

*Comments to Author*: The manuscript entitled “Sulfur-mediated chalcogen versus hydrogen bonds in proteins: a seesaw effect in the conformational space” is nice piece of work. However, the following corrections/clarifications are needed before its publication.

1. Several important references are missing in the introduction part while discussing non-covalent interactions with sulfur (hydrogen bonding and chalcogen bonding). A few noticeable references are J. Phys. Chem. Lett. 2015, 8, 1385-1389, J. Am. Chem. Soc. 2017, 139, 1842−1855, Chem. Sci. 2017, 8, 2667−2670, Acc. Chem. Res. 2020, 53, 8, 1580-1592, J. Mol. Struct. 2020, 1211, 128080, Chem. Soc. Rev., 2022,51, 4261-4286. These references need to be cited and discussed in the revised manuscript.

2. Some of the important parameters such as lower electronegativity, charge and higher polarizability of sulfur should not be ignored while discussing hydrogen bonding with sulfur.

3. Why have different basis sets been used to optimize the monomer and dimer?

4. Why have the criteria for S---H bond distance taken less than 2.8 Å, whereas their sum of van der Walls radii is 3Å? Also, the S---N bond distance criteria were different for CSD (3.35 Å) and PDB (3.6 Å) search. The authors should provide a logical explanation for this.

5. A more detailed explanation should be given in the context of negligible sigma-hole formation on S when it is chelated to a metal center.

6. In the PDB analysis (page 8), the authors mentioned that coordinate files greater than 1 MB in size were excluded from both sets. What is the reason behind choosing based on file size?

7. Fluorescence-based studies:

Why are the experiments done at pH 7.6, slightly higher than the average body pH? Is there any particular reason for this?

The emission maximum for isolated tryptophan in a buffer medium and a protein environment is 350 nm±10 nm. However, in this report, it is 330 nm. Why such a blue shift in the tryptophan emission for MetRS?

For curiosity, can the author explain why fluorescence intensity changes upon ligand binding?

The Kd units in Figures 6C and G need to be correctly written for consistency.

Also, it would be more informative for the readers if the authors provided the emission spectra for titration of proteins and ligands in the supporting information.

8. There is no discussion on CD experiments in the manuscript’s main text. Instead of supplementary 10, the authors can write Figure S10 while discussing it. (page 23)

9. Did the authors perform ITC experiments for all the studied protein-ligand complexes like fluorescence experiments or only for MetRS:Met and MetRS:Nle ? For MetRS:Met complex, the enthalpy contribution is 40%, whereas the entropy contribution is 60% of the total free energy. However, the authors concluded this as prominently entropy-driven; how? Also, how the hydrophobic interactions are responsible for the entropy-driven process?

---

## [Reviewer Report]

*Comments to Author*: The contribution by Adhav et al. represents a very comprehensive and thorough investigation of chalcogen bonds in proteins well worth publishing.

As a general comment, the manuscript has obviously been in the making for quite some time, indicated for example by the CSD searches being executed on a rather old version of the database. One example, I searched for the F1 fragment in Figure 1 B and found 9435 fragment pairs rather than 7854 as indicated, with Nc going from 4.7 % as indicated to 6.1 %. I do not think more updated data would change anything important as far as the major results are concerned, but thought it worthwhile to mention.

Second, after reading the text, although the English language is fine, I sometimes get the feeling pieces of text occur more than once, so I encourage the authors the see if they can clean up the text in certain places and shorten it down. In particular, references to the illustrations are extremely abundant; this could potentially have been done in a more efficient way, i.e. by merging several references to the same illustration.

My only scientific concern deals with the CSD searches. Looking at the structures I retrieved searching for the F1 fragment (see above), I noted that a very large fraction has O-H…S contacts as part of five-membered rings (see attached). As indicated for proteins on page 8, this would have a profound effect on the geometry of the interaction. I thus calculated phi/delta scatterplots for five-membered rings and other rings as well as intermolecular contacts, and they are indeed different (see attached). This would suggest that the selected ranges for δ may lie somewhat to low. Since there is nevertheless a clear difference between hydrogen bonds and chalcogen bonds, it does not impact the way phi and (in particular) delta values are used to distinguish between the two groups, but I nevertheless think this should be addressed in a revised version of the manuscript.

Other than this, I have only a series of minor comments and suggestions.

Page 6, line 3-6. The equation defines ΔE*AB and the text then continues with a discussion of ΔEs, which is evidently corrected for BSSE. Is it really useful to operate with both ΔE*AB and ΔEs, or could BSSE just be included throughout (with this being mentioned, obviously).

Page 6, line 10. MESP is used here for the first time, consider spelling it out (like PES on previous page).

Page 6, line -6: The reference to Figure 4 D here makes no sense.

Page 6, line -4: “CXXXXC” appears here like a Jack-in-a-box with no details provided. A few lines in the Introduction describing the reason for considering this motif would be nice.

Page 7, line -6: “The values for the limits were rounded off to the closest value, which was a multiple of 5 (Supplementary Figure S3 for θ and S4 for δ)”. The reference to Figure S3 is wrong. I assume Figure S4 is correct and that the text refers to the dashed boxes, but this is not really explained.

Page 8, line 3: Not sure I found the division of protein structures into two sets particularly revealing. Consider if you want to keep it.

Page 8, line -3: Reference to the selected 1.9 to 2.8 Å range?

Page 8, line -2: I am not familiar with the use of δ rather than <, i.e. 90° δ θ δ 140° rather than just 90° < θ < 140° as used in Figure 2 A, please explain. The frequent use of the length range notations could be simplified. Also note use on page 13, line -3, which is not clear.

Page 9, line 11-13: As the definitions are given on the top of the page, there is no need to repeat them here.

Page 10, line 7: Here “MetRS” pops out. Those not familiar with this protein need to read to page 21 to learn what this is. Consider adding some information in the introduction, or find another way to handle this.

Page 12, line 8-10: Definitions repeated again.

Page 16, line 10-11: The superscripts on S need to be explained.

Page 19, line 7: A few lines explaining what “negative design” represents in this context?

Page 24, line 14: In my manuscript there is a box in the n-pi* interaction.

Reference list:

I missed a reference to the work by Iwaoka & Babe, Phosphorous, Sulfur, and Silicon, 190:1257-1264, 2015.

Figure 2: Explain boxes. See initial comments on F1.

Figure 3 A. Give the value of the contour surfaces shown?

---

## [Reviewer Report]

*Comments to Author*: Reviewer #1: The manuscript entitled “Sulfur-mediated chalcogen versus hydrogen bonds in proteins: a seesaw effect in the conformational space” is nice piece of work. However, the following corrections/clarifications are needed before its publication.

1. Several important references are missing in the introduction part while discussing non-covalent interactions with sulfur (hydrogen bonding and chalcogen bonding). A few noticeable references are J. Phys. Chem. Lett. 2015, 8, 1385-1389, J. Am. Chem. Soc. 2017, 139, 1842−1855, Chem. Sci. 2017, 8, 2667−2670, Acc. Chem. Res. 2020, 53, 8, 1580-1592, J. Mol. Struct. 2020, 1211, 128080, Chem. Soc. Rev., 2022,51, 4261-4286. These references need to be cited and discussed in the revised manuscript.

2. Some of the important parameters such as lower electronegativity, charge and higher polarizability of sulfur should not be ignored while discussing hydrogen bonding with sulfur.

3. Why have different basis sets been used to optimize the monomer and dimer?

4. Why have the criteria for S---H bond distance taken less than 2.8 Å, whereas their sum of van der Walls radii is 3Å? Also, the S---N bond distance criteria were different for CSD (3.35 Å) and PDB (3.6 Å) search. The authors should provide a logical explanation for this.

5. A more detailed explanation should be given in the context of negligible sigma-hole formation on S when it is chelated to a metal center.

6. In the PDB analysis (page 8), the authors mentioned that coordinate files greater than 1 MB in size were excluded from both sets. What is the reason behind choosing based on file size?

7. Fluorescence-based studies:

Why are the experiments done at pH 7.6, slightly higher than the average body pH? Is there any particular reason for this?

The emission maximum for isolated tryptophan in a buffer medium and a protein environment is 350 nm±10 nm. However, in this report, it is 330 nm. Why such a blue shift in the tryptophan emission for MetRS?

For curiosity, can the author explain why fluorescence intensity changes upon ligand binding?

The Kd units in Figures 6C and G need to be correctly written for consistency.

Also, it would be more informative for the readers if the authors provided the emission spectra for titration of proteins and ligands in the supporting information.

8. There is no discussion on CD experiments in the manuscript’s main text. Instead of supplementary 10, the authors can write Figure S10 while discussing it. (page 23)

9. Did the authors perform ITC experiments for all the studied protein-ligand complexes like fluorescence experiments or only for MetRS:Met and MetRS:Nle ? For MetRS:Met complex, the enthalpy contribution is 40%, whereas the entropy contribution is 60% of the total free energy. However, the authors concluded this as prominently entropy-driven; how? Also, how the hydrophobic interactions are responsible for the entropy-driven process?

Reviewer #3: The contribution by Adhav et al. represents a very comprehensive and thorough investigation of chalcogen bonds in proteins well worth publishing.

As a general comment, the manuscript has obviously been in the making for quite some time, indicated for example by the CSD searches being executed on a rather old version of the database. One example, I searched for the F1 fragment in Figure 1 B and found 9435 fragment pairs rather than 7854 as indicated, with Nc going from 4.7 % as indicated to 6.1 %. I do not think more updated data would change anything important as far as the major results are concerned, but thought it worthwhile to mention.

Second, after reading the text, although the English language is fine, I sometimes get the feeling pieces of text occur more than once, so I encourage the authors the see if they can clean up the text in certain places and shorten it down. In particular, references to the illustrations are extremely abundant; this could potentially have been done in a more efficient way, i.e. by merging several references to the same illustration.

My only scientific concern deals with the CSD searches. Looking at the structures I retrieved searching for the F1 fragment (see above), I noted that a very large fraction has O-H…S contacts as part of five-membered rings (see attached). As indicated for proteins on page 8, this would have a profound effect on the geometry of the interaction. I thus calculated phi/delta scatterplots for five-membered rings and other rings as well as intermolecular contacts, and they are indeed different (see attached). This would suggest that the selected ranges for δ may lie somewhat to low. Since there is nevertheless a clear difference between hydrogen bonds and chalcogen bonds, it does not impact the way phi and (in particular) delta values are used to distinguish between the two groups, but I nevertheless think this should be addressed in a revised version of the manuscript.

Other than this, I have only a series of minor comments and suggestions.

Page 6, line 3-6. The equation defines ΔE*AB and the text then continues with a discussion of ΔEs, which is evidently corrected for BSSE. Is it really useful to operate with both ΔE*AB and ΔEs, or could BSSE just be included throughout (with this being mentioned, obviously).

Page 6, line 10. MESP is used here for the first time, consider spelling it out (like PES on previous page).

Page 6, line -6: The reference to Figure 4 D here makes no sense.

Page 6, line -4: “CXXXXC” appears here like a Jack-in-a-box with no details provided. A few lines in the Introduction describing the reason for considering this motif would be nice.

Page 7, line -6: “The values for the limits were rounded off to the closest value, which was a multiple of 5 (Supplementary Figure S3 for θ and S4 for δ)”. The reference to Figure S3 is wrong. I assume Figure S4 is correct and that the text refers to the dashed boxes, but this is not really explained.

Page 8, line 3: Not sure I found the division of protein structures into two sets particularly revealing. Consider if you want to keep it.

Page 8, line -3: Reference to the selected 1.9 to 2.8 Å range?

Page 8, line -2: I am not familiar with the use of δ rather than <, i.e. 90° δ θ δ 140° rather than just 90° < θ < 140° as used in Figure 2 A, please explain. The frequent use of the length range notations could be simplified. Also note use on page 13, line -3, which is not clear.

Page 9, line 11-13: As the definitions are given on the top of the page, there is no need to repeat them here.

Page 10, line 7: Here “MetRS” pops out. Those not familiar with this protein need to read to page 21 to learn what this is. Consider adding some information in the introduction, or find another way to handle this.

Page 12, line 8-10: Definitions repeated again.

Page 16, line 10-11: The superscripts on S need to be explained.

Page 19, line 7: A few lines explaining what “negative design” represents in this context?

Page 24, line 14: In my manuscript there is a box in the n-pi* interaction.

Reference list:

I missed a reference to the work by Iwaoka & Babe, Phosphorous, Sulfur, and Silicon, 190:1257-1264, 2015.

Figure 2: Explain boxes. See initial comments on F1.

Figure 3 A. Give the value of the contour surfaces shown?

---

## [Reviewer Report]

*Comments to Author*: The authors have addressed the concerns raised by the reviewers in this revised manuscript. The manuscript can be accepted for publication in its current form.

---

## [Reviewer Report]

*Comments to Author*: In the revised version of the contribution by Adhav et al. the authors in my opinion have handled all issues raised by the editor and the two reviewers, including myself, in a satisfactory manner. My comments at this stage are only very minor, with no need for further review.

The English language generally reads very well, but I have some problems with mixing of plural and singular forms in the Abstract in particular. Consider the following corrections:

---

Divalent sulfur (S) formschalcogen bonds (Ch-bonds) via its σ-holes and hydrogen bonds (H-bonds) via its lone pairs. The relevance of these interactions and their interplay for protein structure and function is unclear. Based on an analyses of the crystal structures of small organic/organometallic molecules and proteins and their Molecular Electrostatic Surface Potential (MESP), we show that the reciprocity of the substituent-dependent strengths of the σ-holes and lone pairs correlates with the formation of either Ch-bonds or H-bonds. In proteins, disulfide-linked cysteines preferentially form Ch-bonds, metalchelated cysteines form H-bonds, while methionines form either of them with comparable frequencies. This has implications for the positioning of these residues and their role in protein structure and function. Computational analyses reveal that the S-mediated interactions stabilize protein secondary structures by mechanisms such as helix capping and protecting free β-sheet edges by negative-design. The study highlights the importance of S-mediated Ch-bonds and H-bonds for understanding protein folding and function, development of improved strategies for protein/peptide structure prediction and design, and structure-based drug discovery.

---

If you exclude “(MESP)”, Molecular Electrostatic Surface Potential should not be capitalized, I think. Now this abbreviation is introduced on page 5.

Considering the expression “disulfide-linked cysteines”, I realize this was introduced in response to a comment made by the editor, but personally I don’t like it since “sulfide-linked cysteines” cease to be “cysteines”, they are cystines (a name used only once, in Figure 5, in the current version of the manuscript).

The Results part of the manuscript is where most of the discussion takes place, the Discussion part being used more for concluding remarks. The editor might have a view on this matter.

Minor things:

Page 3, line 10: “… cysteines (Cys-Sγ) and cystines of proteins.”(referring to the above comment)

Page 3, line 12: “Consequently, methionine, cystine and cystine are …”

Page 3, line -1 and -3: Plural form “H-bonds” works better.

Page 3, line -1: Remove final word “bond”.

Page 4, line 7: First part of sentence “Identified in many crystal structures of small, supra, and biomolecules, …” does not make sense.

Page 4, line 16: “…, the precise role of the Ch-bond in protein structures …”

Page 4 line -4: “a H-hond”

Page 4, line -1: Write out AIM on this first occurrence (now done on page 6).

Page 5, line -1: “… in α-helices and β-strands …”

Page 6, line 6: “Fragments F1 - F6 provided in …”

Page 11, line -4: Consider to help the reader a bit by expanding the parenthesis to “(M-S-Y in Figure 4 A)”.

Page 22, legend to Figure 1. Part of the text for (B) obviously belongs to Table 1. The meaning of the asterisk * for F7 - F9 is not explained.

Page 22, legend to Figure 2. For (D) the legend reads “S···O/N contacts”, while in the Figure only “S···O” has been retained. I guess it is wrong in the Figure, as F5 and F6 are shown.

---

## [Reviewer Report]

*Comments to Author*: Editor: I would like to thank the authors for their careful revision. I concur with reviewer #2 that the part most in need of further improvement is the abstract, and am grateful to the great suggestion. Personally, I would leave out any abbreviations from the abstract, though (and in this case decapitalize Molecular Electrostatic Surface Potential.

Regarding the minor suggestions of reviewer #2, it may be helpful to know that negative line numbers correspond to lines counted from the bottom of the page.

Finally, I wish to clarify that I have the same opinion as reviewer #2 regarding cystines. This was a misunderstanding by the authors: I did not ask them to remove cystines from the manuscript, but to carefully check if the correct term was used, as I discovered some that were incorrect (cystines referred to as cysteines or vice versa).

Please implement the final changes, such that we can go ahead publishing the article as soon as possible.

Reviewer #1: The authors have addressed the concerns raised by the reviewers in this revised manuscript. The manuscript can be accepted for publication in its current form.

Reviewer #2: In the revised version of the contribution by Adhav et al. the authors in my opinion have handled all issues raised by the editor and the two reviewers, including myself, in a satisfactory manner. My comments at this stage are only very minor, with no need for further review.

The English language generally reads very well, but I have some problems with mixing of plural and singular forms in the Abstract in particular. Consider the following corrections:

Divalent sulfur (S) formschalcogen bonds (Ch-bonds) via its σ-holes and hydrogen bonds (H-bonds) via its lone pairs. The relevance of these interactions and their interplay for protein structure and function is unclear. Based on an analyses of the crystal structures of small organic/organometallic molecules and proteins and their Molecular Electrostatic Surface Potential (MESP), we show that the reciprocity of the substituent-dependent strengths of the σ-holes and lone pairs correlates with the formation of either Ch-bonds or H-bonds. In proteins, disulfide-linked cysteines preferentially form Ch-bonds, metalchelated cysteines form H-bonds, while methionines form either of them with comparable frequencies. This has implications for the positioning of these residues and their role in protein structure and function. Computational analyses reveal that the S-mediated interactions stabilize protein secondary structures by mechanisms such as helix capping and protecting free β-sheet edges by negative-design. The study highlights the importance of S-mediated Ch-bonds and H-bonds for understanding protein folding and function, development of improved strategies for protein/peptide structure prediction and design, and structure-based drug discovery.

If you exclude “(MESP)”, Molecular Electrostatic Surface Potential should not be capitalized, I think. Now this abbreviation is introduced on page 5.

Considering the expression “disulfide-linked cysteines”, I realize this was introduced in response to a comment made by the editor, but personally I don’t like it since “sulfide-linked cysteines” cease to be “cysteines”, they are cystines (a name used only once, in Figure 5, in the current version of the manuscript).

The Results part of the manuscript is where most of the discussion takes place, the Discussion part being used more for concluding remarks. The editor might have a view on this matter.

Minor things:

Page 3, line 10: “… cysteines (Cys-Sγ) and cystines of proteins.”(referring to the above comment)

Page 3, line 12: “Consequently, methionine, cystine and cystine are …”

Page 3, line -1 and -3: Plural form “H-bonds” works better.

Page 3, line -1: Remove final word “bond”.

Page 4, line 7: First part of sentence “Identified in many crystal structures of small, supra, and biomolecules, …” does not make sense.

Page 4, line 16: “…, the precise role of the Ch-bond in protein structures …”

Page 4, line -1: Write out AIM on this first occurrence (now done on page 6).

Page 5, line -1: “… in α-helices and β-strands …”

Page 6, line 6: “Fragments F1 - F6 provided in …”

Page 11, line -4: Consider to help the reader a bit by expanding the parenthesis to “(M-S-Y in Figure 4 A)”.

Page 22, legend to Figure 1. Part of the text for (B) obviously belongs to Table 1. The meaning of the asterisk * for F7 - F9 is not explained.

Page 22, legend to Figure 2. For (D) the legend reads “S···O/N contacts”, while in the Figure only “S···O” has been retained. I guess it is wrong in the Figure, as F5 and F6 are shown.